# Context Preserving Autoregressive Frame Generation for Bounded Video

## Abstract

Recently, various video generation methods have been proposed, as diffusion models demonstrate their superior ability to generate high-quality videos. Specifically, autoregressive approaches have been suggested to enable the generation of videos of arbitrary length. However, the methods are not suitable for bounded video generation, as they generate open-ended videos. Moreover, recent methods for bounded video generation rely on flipping frames to satisfy the boundary constraint imposed by the ending frame. However, this approach contradicts the inherent bias of video models to generate frames in forward direction, limiting the generation capability. Accordingly, we propose a novel autoregressive approach for bounded video generation. Specifically, we introduce a context-aware bidirectional denoising method that progressively generates frames in both forward and backward directions while considering the frame context. Then, we propose a method to mitigate the context gap between the two directions, to ensure smooth and coherent transition between the sequences. Experimental results demonstrate the superiority of our approach over previous methods. Specifically, as our method aligns with the video model's forward generation bias, the output videos present more realistic motion dynamics. Moreover, our method outputs frames with enhanced visual quality by maintaining a consistent frame length for model input. More results can be found in our project page[1].

## 1 Introduction

Recent advances in video diffusion models have revolutionized the video generation method, achieving remarkable visual quality and creative flexibility conditioned by text. The proposed video diffusion models are based on various architectures such as U-Net[2, 3, 10, 24]and transformers[6, 13, 14]. Both architectures output high-quality videos with enhanced temporal coherence and semantic alignment. However, they generate an entire video sequence at once, treating the temporal axis as an additional dimension of the diffusion latent vector. This significantly increases computational burden, making long video generation challenging. Accordingly, autoregressive approaches have been suggested, which generate video frames progressively. The methods leverage previous frames as a conditioning factor for later frames, which is able to generate videos with flexible length. This approach facilitates long-range dependencies, allowing for consistent motion propagation throughout the sequence[18, 21, 26]. However, training the video model requires extensive computational resources and access to refined video datasets. To overcome the limitation, training-free methods[11, 17, 22] have recently been proposed to enhance long-range dependencies without requiring enormous training resources. Despite their advantages, these methods primarily focus on open-ended video generation, making them unsuitable for bounded video generation.

---

[1]https://sites.google.com/view/cpag-video

Submitted to 39th Conference on Neural Information Processing Systems (NeurIPS 2025). Do not distribute.

Bounded video generation refers to the task of synthesizing a video sequence that adheres to predefined bounds. This task is essential for various applications, including video interpolation, video completion, and content extension while maintaining temporal coherence. Recent methods[5, 23, 25] proposed to fuse forward and reversed frames to satisfy the boundary frame condition. However, existing approaches utilize frame-flipping techniques, where reversed frames are synthesized by flipping the generated forward frames. This introduces inherent limitations, as flipping the output frames encounters the conflict to the model's bias to generate frames in forward direction. For instance, irreversible motions are generated with unrealistic dynamics, since the method flips output frames to obtain time-reversal video sequence. Moreover, the methods have constraints on video length, as they fuse sequences of fixed length. Specifically, the mismatch of input sequence length between the training and inference limits the model's generation capability, as suggested in the prior work [17].

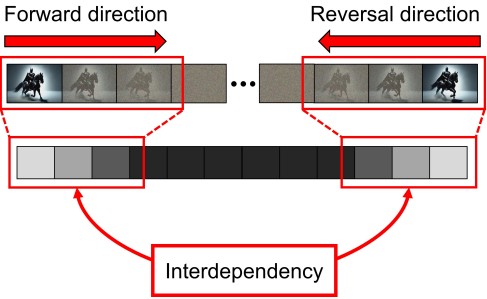

Figure 1: Concept of the proposed bidirectional autoregressive approach, addressing context gap.

To address the limitations, we propose a novel autoregressive approach for bounded video generation. Specifically, as shown in Fig. 1, our method aims to generate coherent bounded video, considering the interdependency between the two sequences. The proposed method consists of two parts. First, we suggest a context-aware bidirectional denoising method. Specifically, inspired by Stein Variational Gradient Descent (SVGD), we present context gradient that exchanges score information with neighboring frames. Then, we denoise frames by the diffusion steps, scheduled for bidirectional generation. Second, we propose a method to reduce the context gap between the two directions. To be specific, a content generated in each direction may diverge unless the interdependency between the two sequences is explicitly addressed. Accordingly, we mitigate the context gap by the proposed frame initialization strategy and the mixing of context gradients. Consequently, our method effectively guides diffusion process to generate videos with improved consistency while satisfying boundary conditions. Compared to previous methods based on the frame-flipping technique, our method aligns with the model's bias when generating frames in reversal direction. As a result, our method generates videos with more realistic dynamics. Fig. 2 summarizes a main conceptual difference of our work compared to the previous approach. Our contributions are summarized as follow:

- We suggest a novel bidirectional autoregressive approach for bounded video generation, which aligns with the video model's forward generation bias.

- We propose a context gradient which enables to exchange the score information with neighbor frames, and present a context-aware bidirectional denoising method.

- We explore the context gap between the two directions, and propose a novel method to mitigate the context gap to generate coherent videos.

- We empirically demonstrate the proposed method, highlighting the effectiveness of our autoregressive approach for bounded video generation.

## 2 Related works

### 2.1 Video diffusion model

Recently, many research work consistently demonstrate the superior ability of the diffusion model for video generation. [2, 3, 6, 14]. Moreover, to address the growing demand for long video generation, various methods have been proposed, including autoregressive approach that enables the generation of videos of arbitrary length. Specifically, [21, 26] proposed to obtain autoregressive models by training with designed objective. However, by the excessive resources for training models, training-free methods have been proposed. Gen-L-Video[22] proposes temporal co-denoising, which uses the average of multiple predictions to generate one frame. FreeNoise[17] suggests a novel temporal attention matrix to retain long-range correlation. FIFO[11] proposes iterative diagonal denoising to generate video frames in autoregressive manner. Specifically, FIFO suggests latent partitioning

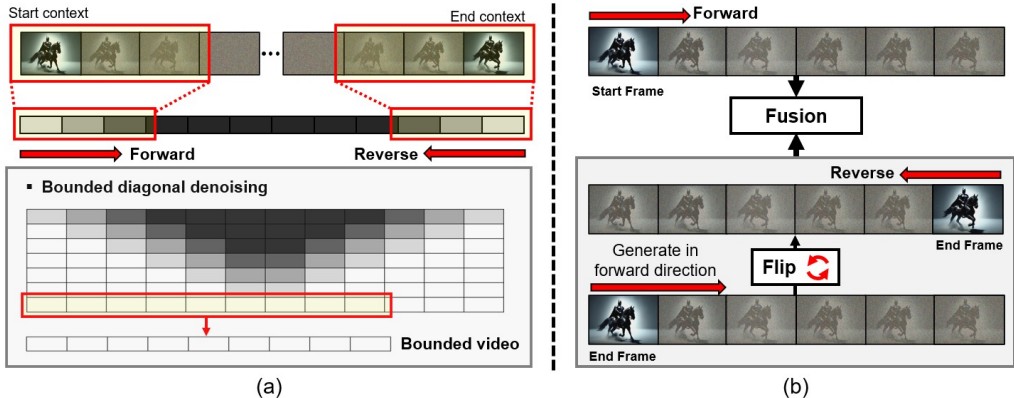

Figure 2: Comparison of our method with previous approach. (a) Proposed autoregressive approach. (b) Previous methods based on flipping frames to generate reversal sequence.

and lookahead denoising to reduce the training-inference gap. However, previous methods generate open-ended videos, which is not suitable for bounded video generation. In this paper, we propose a novel autoregressive method for bounded video generation, which satisfies the boundary conditions while allowing the generation of video in arbitrary length.

## 2.2 Bounded video generation

With the rise of video diffusion models, various methods have been proposed for bounded video generation. Generative Inbetweening[23] fine-tunes the projection matrix, and interpolate forward and reversed frames. Specifically, it generates reversed frames by rotating the self-attention matrix and flipping the output frames. However, to avoid the computational cost of fine-tuning, training-free methods have emerged. TRF[5] obtains reversed frames by flipping the generated outputs, and linearly interpolates with forward frames. Moreover, authors suggest noise injection for smooth connectivity. ViBiDSmapler[25] proposed a novel diffusion denoising step that combines forward and reversal frame generation process. Specifically, the method denoises the frames in forward direction, and flips the frames to predict noise in reversal direction. Moreover, DDS guidelines[4] is applied to enforce the boundary condition. However, video models are inherently biased to generate frames in forward temporal direction, making such approaches misaligned with the model's natural behavior. Moreover, they fuse the sequences with fixed length, which constraints the flexibility of video length. As introduced in [17], the difference of input frame length degrades the model's generation capability. To address the limitation, we propose an autoregressive approach to generate reversed frames by adding more noise in desired direction. Moreover, our method enables to generate videos of arbitrary length with consistent input frame length, which is restricted in the previous methods.

## 3 Main Contribution

### 3.1 Key observation: Autoregressive reversed frame generation

Previous methods[5, 23, 25] synthesize reversed videos by flipping frames generated in forward direction, which is counter to the model's bias. To address the issue, we approach the generation of reversed frame in autoregressive generation. Specifically, inspired by the diagonal denoising[11, 18], we explore a reversed diagonal denoising as shown in Fig. 3, which applies more noise in the preceding frames. This provides an information of future frames to the past frames, enabling the generation of reversed frame without opposing the model's inherent forward-generation bias.

The results demonstrate that the method successfully generates the reversed frames. As shown in Fig. 3,

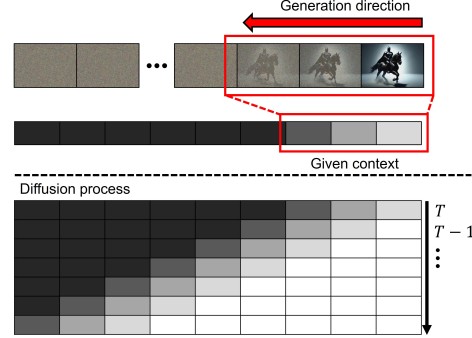

Figure 3: Diffusion time of reversed diagonal denoising. Dark colors refer high noise levels.

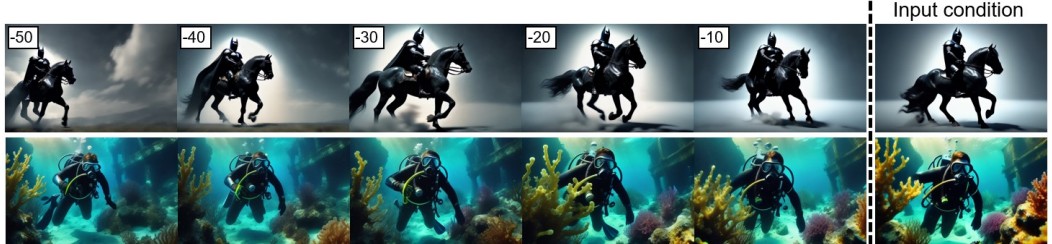

Figure 4: Reversed frame generation by the reversed diagonal denoising. The negative numbers indicate the difference in frame index from the input image.

126  the motion of the running horse is irreversible, requiring the movement of its legs to align with the
127  running direction. The method generates a natural motion of a running horse, while satisfying the
128  boundary condition of the end frame. Similarly, the other irreversible motions are generated with
129  realistic dynamics, such as air bubbles rising from a scuba diver. Based on this observation, we
130  propose a novel autoregressive approach for bounded video generation, which aligns with the video
131  model's forward-generation bias. More experimental details are provided in Appendix.

## 3.2  Main method

### 3.2.1  Context-aware bidirectional autoregressive denoising

134  We introduce context-aware denoising method,
135  inspired by Stein Variational Gradient De-
136  scent (SVGD) [12, 15]. Specifically, SVGD
137  provides the gradient for particle $x$ to approximate
138  target distribution $p$ as follows:

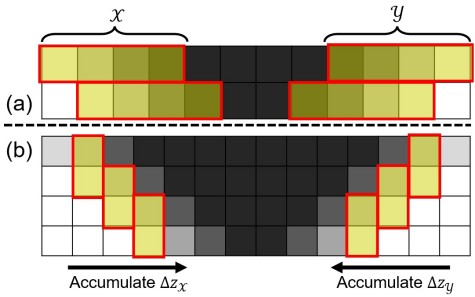

$$\Delta x = \mathbf{E}_{x' \sim q(x')}[k(x, x')\nabla_{x'} \log p(x') + \nabla_{x'} k(x, x')] \tag{1}$$

139  For known distribution $q$, and positive definite ker-
140  nel $k$. The SVGD provides a direction toward high-
141  density region using the other particles $x'$, preventing
142  collapse by pushing each other with the second term.

Figure 5: (a) Update of index set $\mathcal{X}, \mathcal{Y}$. (b) Accumulation of the gradient $\Delta z_{\mathcal{X}}, \Delta z_{\mathcal{Y}}$

143  We suggest context-aware gradient, inspired by
144  SVGD. For the video generation, closely located frames should exchange the score update more than
145  the far one. Accordingly, we set $q$ as the distribution of preceding frames in the same diffusion time
146  step. Moreover, we exclude the second term, as the video model already prevents the collapse. For
147  the DDIM where $z_t = \sqrt{\alpha_t}\hat{z} + \sqrt{1 - \alpha_t}\epsilon$, we present the gradient for forward direction, $\Delta z_{\mathcal{X}}$, in
148  clean manifold as follows:

$$\Delta z_{\mathcal{X}}(i) := \sum_{j<i} g(i, j)\Delta \hat{z}_j \tag{2}$$

149  where $i$ and $j$ are frame indices, $g$ is monotonic decreasing function for the frame distance, and $\Delta \hat{z}_j$
150  is the predicted update of preceding frames $z_j$ in clean manifold. The gradient for reverse direction
151  $\Delta z_{\mathcal{Y}}$ is defined in similar way. For implementation, we update the gradients by moving average
152  that satisfies $g$, as shown in Fig. 5. Thereby, $\Delta \hat{z}$ captures contextual information for each time step.
153  Additionally, we apply DDS[25] to the frame for input context, to satisfy the boundary condition.

154  Then, we set time schedule to generate frames in both directions in autoregressive manner. As
155  illustrated in Fig. 5, we add more noise in the direction of the generation, which is inspired by
156  previous method[11, 18]. Specifically, the model $\phi$ denoises the sequence $\{z_f\}_{f=1}^F$, where each
157  frame $z_f$ has different diffusion time $\tau(f)$ as follows:

$$\tau(f) = \max\{f + (T - l_{CTX}), T\} \tag{3}$$

$$z_{f,\tau-1} = \phi(\{z_{f,\tau}\}_{f=1}^F, \{\tau(f)\}_{f=1}^F) \tag{4}$$

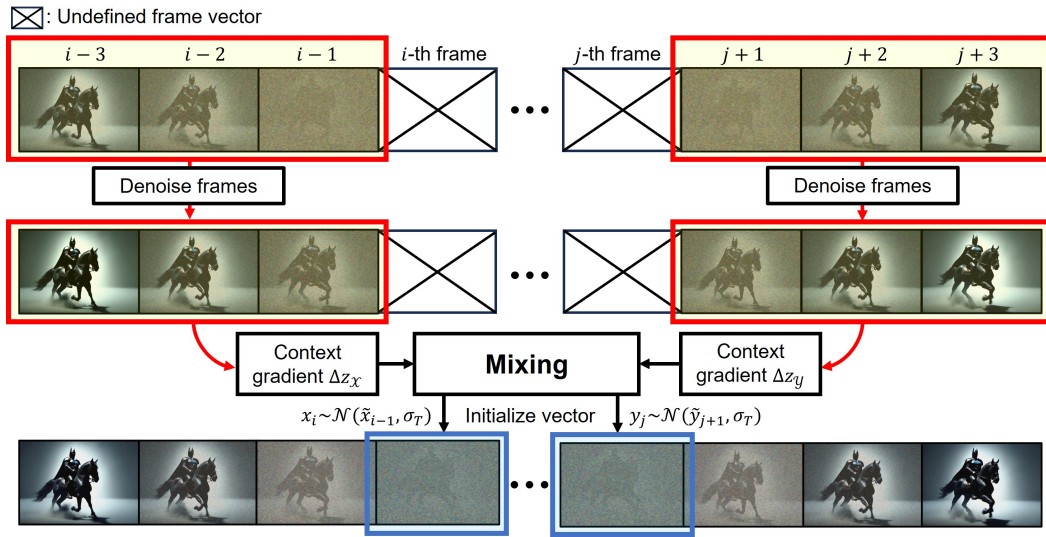

Figure 6: Overall diagram of proposed context-aware bidirectional generation. We extract the context gradient $\Delta z_{\mathcal{X}}, \Delta z_{\mathcal{Y}}$ from each sequence, and utilize them to generate the new frames.

where $f$ is frame index, $F$ is input frame length of model, $l_{CTX}$ is the length of input context, and $T$ is maximum number of time. Here, we denote a set of indices for each sequence as $\mathcal{X}$ and $\mathcal{Y}$, respectively. $\mathcal{X}$ and $\mathcal{Y}$ are updated for each denoising step by removing the index of fully denoised frame and adding the index of new frame. If there are no new frames to add, we apply lookahead denoising[11]. We provide details for the denoising process in the Appendix.

### 3.2.2 Reducing context gap for coherent video

In the proposed bidirectional generation method, main challenge becomes to address context gap between the forward and reversed frames. Specifically, as each direction retains its own frame context, discrete transition emerges in the middle when the two sequences are combined to generate bounded video. Hence, the interdependency between the two directions should be considered to generate coherent video. Accordingly, we propose a method for a smooth transition between the two sequences. Specifically, we propose the method for frame initialization, and the context gradient to guide frames to consider the context in global perspective. For clarity, in this section, we denote the diffusion vector $z$ as $x$ for forward direction, and $y$ for the backward direction. First, we design the mixing ratio $R(i, j)$ which determines the ratio of information exchange between the two sequences in global view. Similarly, we define the frame kernel $k(i, j)$ to consider local context. Then, we initialize $x_i$ using the predicted of frames in set $\mathcal{X}, \mathcal{Y}$, as follows:

$$\tilde{x}_{i-1} = (1 - r) \sum_{m \in \mathcal{X}} k(i, m)\hat{x}_m + r \sum_{n \in \mathcal{Y}} k(i, n)\hat{y}_n \tag{5}$$

$$x_i \sim \mathcal{N}(\tilde{x}_{i-1}, \sigma_T) \tag{6}$$

for maximum noise level $\sigma_T$, mixing ratio $r = R(i, j)$ and frame kernel $k$. Likewise, $\hat{y}_j$ is sampled in similar way. It is simple, but effective, as the interpolation in the noisy manifold results the semantic interpolation in clean manifold[8, 16, 19]. Next, we suggest a context gradient which provides a guidance considering the global context. Following the linearity of equation (2), we obtain the context gradient as:

$$\Delta \tilde{x}_i = (1 - r)\Delta z_{\mathcal{X}} + r\Delta z_{\mathcal{Y}} \tag{7}$$

$$\Delta \tilde{y}_j = (1 - r)\Delta z_{\mathcal{Y}} + r\Delta z_{\mathcal{X}} \tag{8}$$

which provides guidance to each frame during diffusion process. To clarify our method, we present pseudo code in Algorithm 1.

---

**Algorithm 1** Autoregressive generation for bounded video

---

**Input:** Forward set $\mathcal{X}$, reverse set $\mathcal{Y}$, frame latents $\{z_f\}_{f=0}^{L}$

**Output:** Denoised latents $\{z_f\}_{f=1}^{L-1}$, updated $\mathcal{X}, \mathcal{Y}$

// Bounded diagonal desnoising //

1: Get time steps $\{\tau_f\}_{f=0}^{L}$
2: For $i \in \mathcal{X}$, predict $\{\hat{z}_i\} = \phi(\{z_i\}, \{\tau(i)\})$ using $\Delta \tilde{z}_i$
3: For $j \in \mathcal{Y}$, predict $\{\hat{z}_j\} = \phi(\{z_j\}, \{\tau(j)\})$ using $\Delta \tilde{z}_j$
4: Update $\tau(f)$

// Initialization of next frames $i'$, $j'$ //

5: Get $c_{\mathcal{X}} = \sum_{m \in \mathcal{X}} k(i', m)\hat{z}_m$, $c_{\mathcal{Y}} = \sum_{n \in \mathcal{Y}} k(j', n)\hat{z}_n$ and $r = R(i', j')$
6: $\tilde{z}_{i'-1} = (1 - r)\, c_{\mathcal{X}} + r\, c_{\mathcal{Y}}$
7: $\tilde{z}_{j'+1} = (1 - r)\, c_{\mathcal{Y}} + r\, c_{\mathcal{X}}$
8: Sample $z_{i'} \sim \mathcal{N}(\tilde{z}_{i'-1}, \sigma_T)$ and $z_{j'} \sim \mathcal{N}(\tilde{z}_{j'+1}, \sigma_T)$
9: Update gradients $\Delta z_{\mathcal{X}}, \Delta z_{\mathcal{Y}}$ by moving average
10: Add new indices: $\mathcal{X}.\text{add}(i')$, $\mathcal{Y}.\text{add}(j')$
11: **if** $\min \tau(f) = 0$ **then**
12:     Output frames: $z_0$, $z_L$
13:     Delete indices: $\mathcal{X}.\text{remove}(0)$, $\mathcal{Y}.\text{remove}(L)$

---

# 4 Experimental Results

## 4.1 Implementation Details

Our method requires an initial set of frames for input context, as we progressively generates video frames. In our experiment, we generate a short video, and use it as an initial context. Our code is based on the official implementation of FIFO [2]. We compare our method with previous training-free methods. First, we compare our method with FIFO[11], a state-of-art autoregressive method. Then, we select TRF[5] and ViBiDSampler[25] which are recent methods for bounded video generation. Specifically, we adjust the input frame length for TRF and ViBiDSampler, as the methods generates whole frames at once, which are not the autoregressive manner. For the experiment of the bounded video generation, we conducted experiments for two tasks, where the first task is generation of looped video by identical bound, and the second task is the generation of smooth video by the two bounds[5]. We randomly create 200 prompts by GPT-4[1] for each task. We utilized the publicly release videocrafter2 [3] model for the experiments. The experiments are conducted by H100 GPUs. For the comparison, we evaluate the videos in three aspects, temporal consistency, visual quality and boundary satisfaction. We selected Temporal Flickering, Background Consistency in VBench [9] to evaluate temporal smoothness, and FVD[20] and FID[7] to compare perceptual quality of frames. For boundary satisfaction, we calculate mean absolute error (MAE) of frames in pixel space. More details can be found in the Appendix.

## 4.2 Bounded video generation

We present the result of bounded video generation in Fig. 7, which demonstrates the superiority of our method compared to the previous methods. First, for the experiment on identical bound, our method successfully generates the looping video with high visual quality, while satisfying the input bounds. Specifically, our method generates the detailed and realistic motion, such as splashing water and natural movement of heron flapping the wings. In case of FIFO[11], the method results open-ended video, which fails to satisfy the bound. In contrast, TRF[5] generates bounded videos following the input bound. However, the output shows blurry texture and motion. Specifically, the texture of splashing water and the motion of the wings become overly smooth. ViBiDSampler[25] outputs finer details than TRF, but still blurry compared to our method. Here, the degradation in visual quality arises from a mismatch of input sequence length used for training and inference, also shown in previous work[17]. In contrast, our approach maintains the consistent length by autoregressive approach, effectively avoiding the degradation.

---

[2]https://github.com/jjihwan/FIFO-Diffusion_public

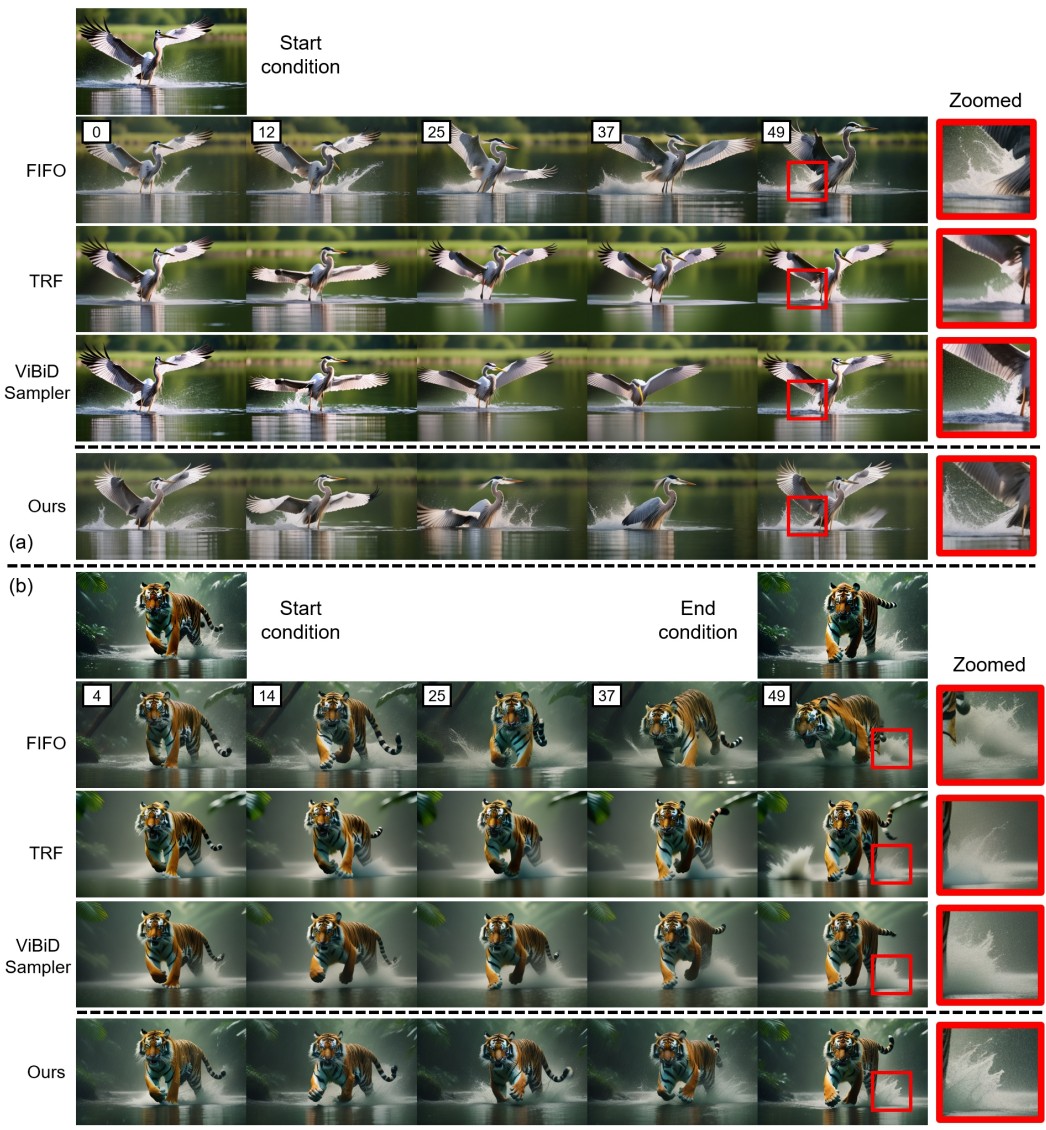

Figure 7: Main result of bounded video generation. (a) Looped video generation by identical bound. (b) Bounded video generation conditioned on two bounds, distinct start and end frames.

Table 1: Quantitative comparison for bounded video generation. Our method successfully generate bounded video with enhanced visual quality.

| | Temporal Consistency | | Visual quality | | Boundary satisfaction | | |
|---|---|---|---|---|---|---|---|
| | Temporal Flickering ↑ | Background Consistency ↑ | $FVD_{16} \downarrow$ | FID ↓ | MAE(Loop) | MAE(Start) | MAE(End) |
| TRF | 0.957 | 0.966 | 591.44 | 52.75 | 0.066 | 0.137 | 0.122 |
| ViBiD | 0.952 | 0.967 | 429.28 | 35.59 | 0.074 | 0.139 | 0.114 |
| FIFO | 0.934 | 0.961 | 386.72 | 30.28 | 0.202 | 0.152 | 0.235 |
| **Ours** | **0.957** | **0.969** | **393.76** | **31.61** | **0.076** | **0.097** | **0.101** |

Second, for the experiment for two bounds, our method also outperforms the other method in terms both of visual quality and boundary satisfaction. Our method shows detailed texture for both object and background, specifically the texture of leaves in jungle and the splashing water. Moreover, our method successfully generate frames in between two bounds with enhanced temporal consistency. On the contrary, FIFO fails to satisfy the ending bound, as it generates frames autoregressively without any constraints. In case of TRF, the result satisfies the bounds, maintaining the temporal coherent

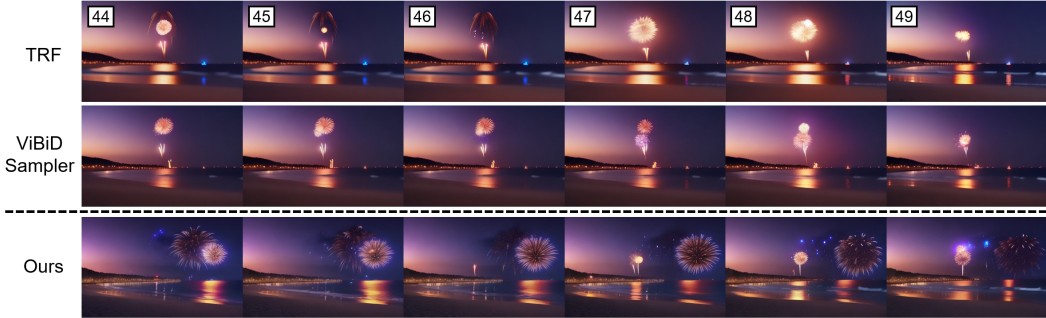

Figure 8: Comparison of irreversible motions in single bounded video with 50 frames. Numbers in the images indicate frame index. We present a typical irreversible motion, movement firework flares.

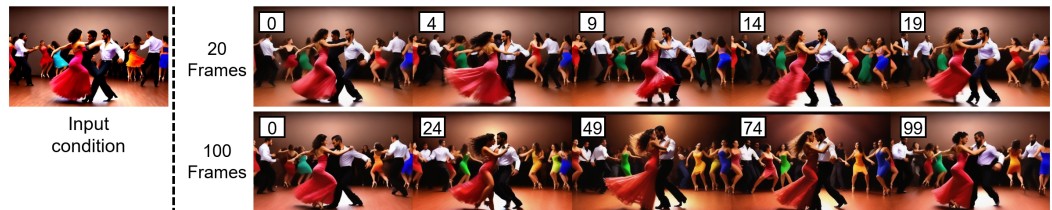

Figure 9: Ablation study for video length. We observe the deviation of context according to the length of video.

between the frames. However, the frames demonstrate low visual quality with smoothed texture, which is coherent result for looped video generation. Moreover, the method fails to generate the realistic motion of tiger. Likewise, ViBiDSampler results frames with better visual quality and motion than TRF, but worse than the proposed method. Both previous methods suffers from degraded visual quality due to the difference in input sequence length between training and inference. In contrast, the proposed autoregressive approach avoids the degradation by allowing the consistent input length. Overall, the results in both experiments demonstrates the superiority of our method.

Further, we provide quantitative comparison in Table 1. Compared to FIFO, our method outputs enhanced temporal consistency by considering the frame context. Moreover, boundary conditions are satisfied in our method, which is not in FIFO. Compared to TRF and ViBiDSampler, our method outputs better visual quality and boundary satisfaction.

### 4.3 Discussion

#### 4.3.1 Irreversible motion

Our method leverages the video model's bias when generating the reversed frames, whereas the previous methods are counter to the bias by flipping the frame. To support the claim, we present empirical evidence in Fig. 8. Specifically, we compare the frames near the end of video with irreversible motion, the flares in firework. Our method successfully generates the natural fading of firework flares. Whereas, TRF and ViBiDSampler output unnatural shrinkage of the flares, which are reversed dynamics. The results demonstrate the superiority of the proposed approach in generating realistic motions compared to the previous approach.

#### 4.3.2 Video length

Our method generates frames in an autoregressive manner, enabling the generation of bounded videos of arbitrary length. However, as the video length increases, the generated frames gradually deviate from initial boundary frames. Accordingly, we further explore the deviation of context as the length of video increases. As shown in Fig. 9, we conduct the ablation study on video lengths for looped video generation. In the case of a short video with 20 frames, the generated frames maintain the initial context of input image, preserving the background and overall scene. However, for a longer video with 100 frames, a gradual shift in context is observed. Specifically, as the frames progress, the dancing couple dominates the scene where the stage gradually changes with the emerge of focused

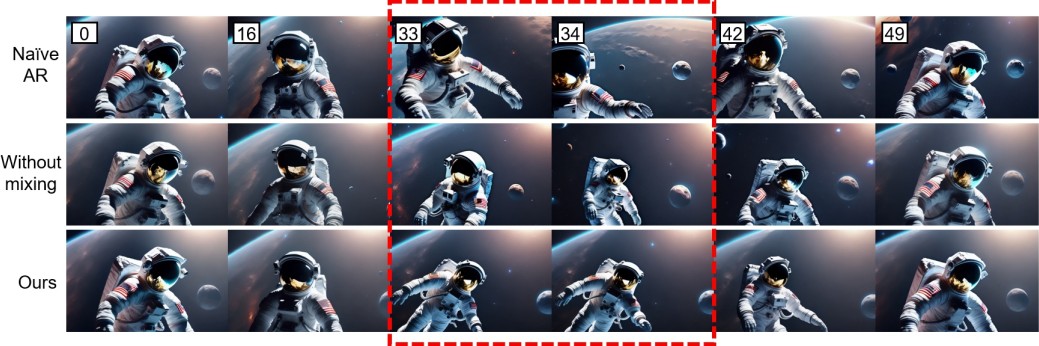

Figure 10: Output frames to explore context gap. Abrupt change between intermediate frames is observed in the middle of video caused by the context gap. Red rectangle highlights the discontinuity.

lighting. Despite the deviation, the frames gradually reconstructs the original scene in the end. The results demonstrate that generation of long video accompanies the contextual shifts by the weakened long-range dependency. Additionally, it is notable that our method successfully generates a coherent bounded video even in the presence of the deviation.

### 4.3.3 Context gap

To evaluate the effectiveness of the proposed method, we conduct an ablation study exploring the context gap in the output video with single bound. We compare our method with two variants, one without consideration of global context ('Without mixing') and another using a naive combination of diffusion time schedules for forward and backward generation ('Naive AR'). Compared to Naive AR, the consideration of context improves temporal consistency in both directions, as shown in Fig. 10. However, a discontinuity emerges in the middle, where the transition occurs between the sequences. This is caused by the context gap of the two directions, leading to abrupt changes in motion and background. In contrast, our method successfully resolves the discontinuity, showing the consistent video frames.

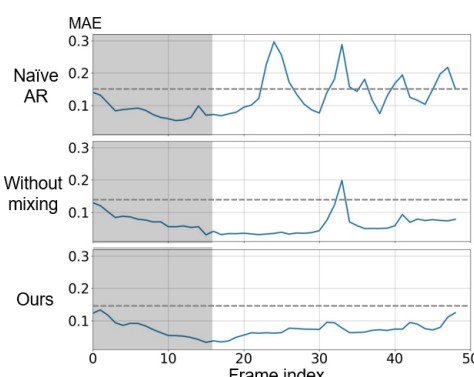

Figure 11: MAE between frames. Gray region indicates the input bound frames.

For the quantitative comparison, we measure mean absolute error (MAE) between intermediate frames. As shown in Fig. 11, three videos shows similar trends of MAE in gray region, as they follow the given bound. However, each method shows distinct tendency after the bound. Considering the context enhances the temporal consistency, lowering the MAE. However, a large peak emerges in the middle, which indicates the context gap between the two directions. When the mixing is applied, the peak is removed, which indicates a smooth transition. The results demonstrate that our method successfully outputs coherent bounded video, effectively mitigating the context gap.

## 5 Conclusion

We proposed a novel bidirectional autoregressive approach for bounded video generation. First, we suggest context-aware autoregressive denoising which generates frames in both forward and reversed directions, effectively leveraging the video model's forward generation bias. Moreover, we introduce the context gradient to capture the contextual information of frames. Then, we propose a novel method, to reduce the context gap between the two directions. Our method is capable of generating bounded videos of arbitrary length while satisfying the boundary conditions. Experimental results verify the outperformance of our method compared to the related methods. Moreover, we present inherent limitation of previous approach by comparing the results for irreversible motion. We further conduct the ablation studies to validate the effectiveness of each component in our method.

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

# A  Additional results

We provide additional results for both identical bound and dynamic bound in Fig. 12 and Fig. 13. Our method successfully generates the bounded video with high quality while satisfying the boundary condition.

# B  Experimental settings

## B.1  Key observation

We present autoregressive reversed frame generation as key observation in the main paper. The implementation of the experiment is based on the publicly released code of FIFO[3], which generates open-ended video in forward direction by autoregressive manner. Accordingly, the hyperparameters are identical to the baseline. For the experiment to generate the reverse direction, we reversed two key components in FIFO[11], the time schedule and a direction of lookahead denoising. We utilized publicly released video model, videocrafter2[3]. The experiments are conducted by H100 GPUs. The text conditions for the Figure 4 in main paper are "A dark knight riding horse in glass land" and "scubar diver exploring the shipwreck underwater", which are selected to visualize the successful generation of irreversible motions in reversed direction.

## B.2  Proposed method

We follow the experimental setting of TRF[5] which suggests three categories for bounded video generation task. Specifically, for single bound task, we generate a video conditioned on a text prompt and set as the bound. For two bound task, following 'Dynamic Bound' in TRF, we sample frames from the generated video with moving object. For the prompts in Fig. 7 in main paper, we used "Cinematic photo of a heron taking off in slow motion from still water" for single bound, and "Cinematic photo of a tiger charging through dense jungle in a monsoon, leaves and water flying with each step" for two bounds.

**Hyperparameters**
Our implementation is based on implementation of FIFO. Hence, we share the hyperparameters and settings with the baseline. DDIM[19] schedule with 64 inference steps is utilized. The length of videos is 50 frames, unless specified. We defined the frame kernel $k(i, j)$ and mixing ratio $R(i, j)$ as gaussian kernel. Specifically, each function is defined as follows:

$$k(i, j) = c_k \exp\{-(i - j)^2/\gamma_k\} \tag{9}$$

$$R(i, j) = \lambda_R c_R \exp\{-(i - j)^2/L\gamma_R\} \tag{10}$$

where $\gamma_k, \gamma_R, \lambda_R$ are the hyperparameter, $L$ is the total length of video sequence. $c_k$ and $c_R$ are the normalizing constant. In our method, each parameters are set as $\gamma_k = 16, \gamma_R = 4, \lambda_R = 0.5$. Specifically, $k(i, j)$ has a length of 16 and is normalized so that the sum of each element becomes one. The weight to update $\Delta z$ by moving average is set as 0.6 for single bound, and 0.8 for two bounds.

**Evaluation metric**
We evaluate videos in three different perspective, which are temporal consistency, visual quality of frames and satisfaction of the given boundary. To evaluate the temporal consistency, we utilize the temporal flickering and background consistency suggested in VBench[9], which calculate the difference between frames in mean absolute error(MAE) using pixel values and CLIP features, respectively. Specifically, in both metrics, higher value indicates more smoothness in video, where the maximum value is 1.

To evaluate the visual quality, we present $FVD_{16}$[17, 20] and FID[7]. Specifically, following the previous work[17], we compare the output with input context of 16 frames, sampling the 16 frames in the generated video. For FID, we calculate the metric between the input bound frames and the output frames. We calculate the metric using 400 generated videos, where the prompts are randomly generated by GPT-4[1].

---

[3]https://github.com/jjihwan/FIFO-Diffusion_public

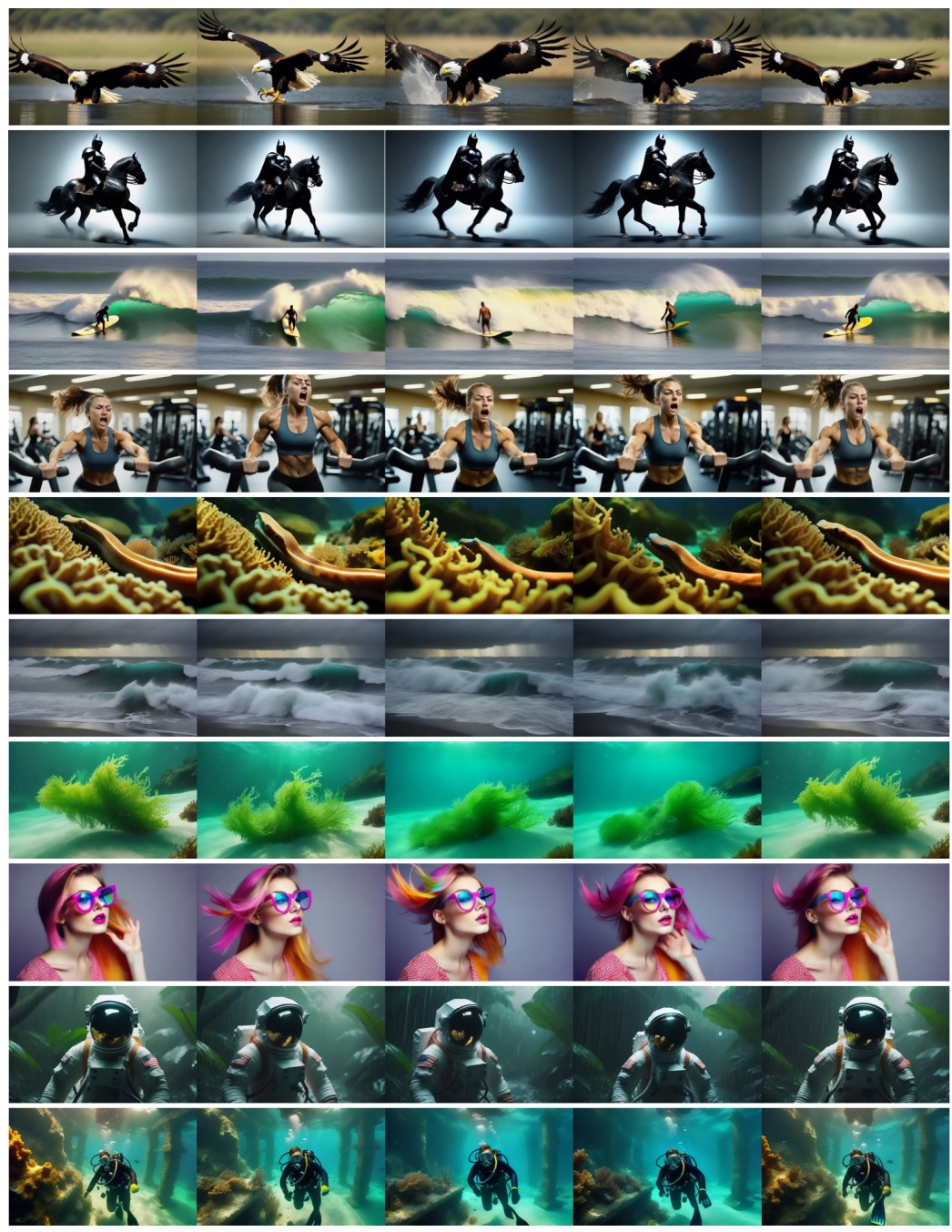

Figure 12: Additional result for looped video with identical bound.The selected frame indices are {0, 12, 25, 37, 49}

To evaluate the boundary satisfaction, we simply calculated the MAE difference of frames in pixel space. Specifically, for the looped video with single bound, we calculate the MAE between the first frame and the last frame. In case of bounded video with two different bounds, we calculate the MAE for the start condition, as the difference between the last context frame and the first generated frame. For instance, the last context frame is 16 and the other one is 17, for the input context length of 16 in the generated video. Then, we obtain the MAE for end condition, as the difference between the last frame of generated video and the first frame of the end context. The values are normalized with maximum pixel value, 255. Lower value indicates better boundary satisfaction.

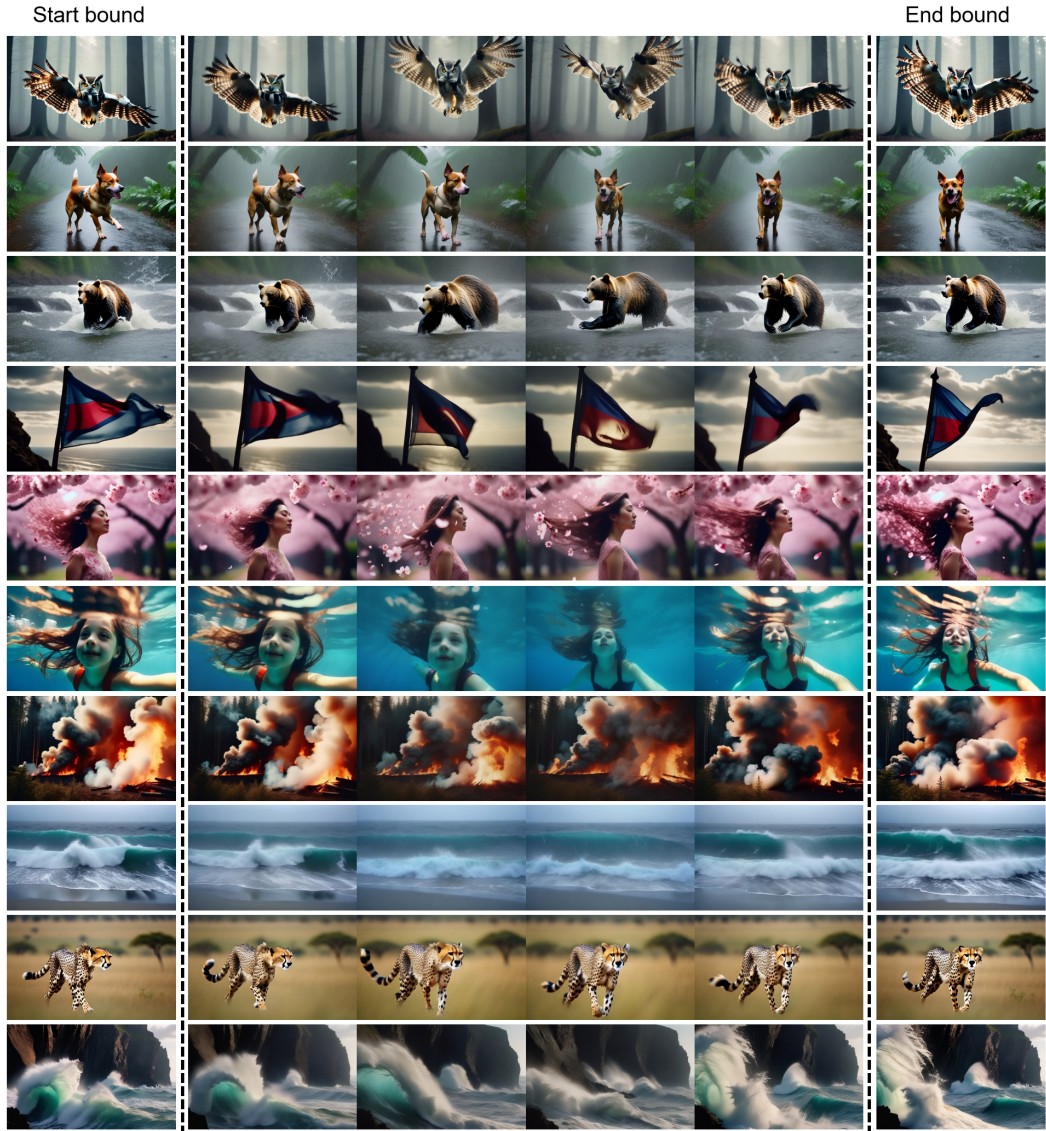

Start bound                                                                End bound

Figure 13: Additional result for looped video with dynamic bound. The selected frame indices are {4, 19, 34, 49}

## C  Denoising process

### C.1  Inference stages

Our method has three different phases for video generation, since the sequence length gradually shortens as generated frames are removed from both sides. We provide a detailed explanation of each phase.

**Phase 1: No overlaps between $\mathcal{X}$ and $\mathcal{Y}$**

In this phase, we consistently add new frames to each set, $\mathcal{X}$ and $\mathcal{Y}$. The new frames are initialized by the mixing of frame predictions, as described in the main paper. If the size of $\mathcal{X}$ and $\mathcal{Y}$ exceed the model's input frame length, lookahead denoising is applied independently to each side. We do not alternate the direction of lookahead denoising in this phase.

**Phase 2: No more new frames**

In this phase, there is no more new frame to initialize. Now, we apply alternating lookahead

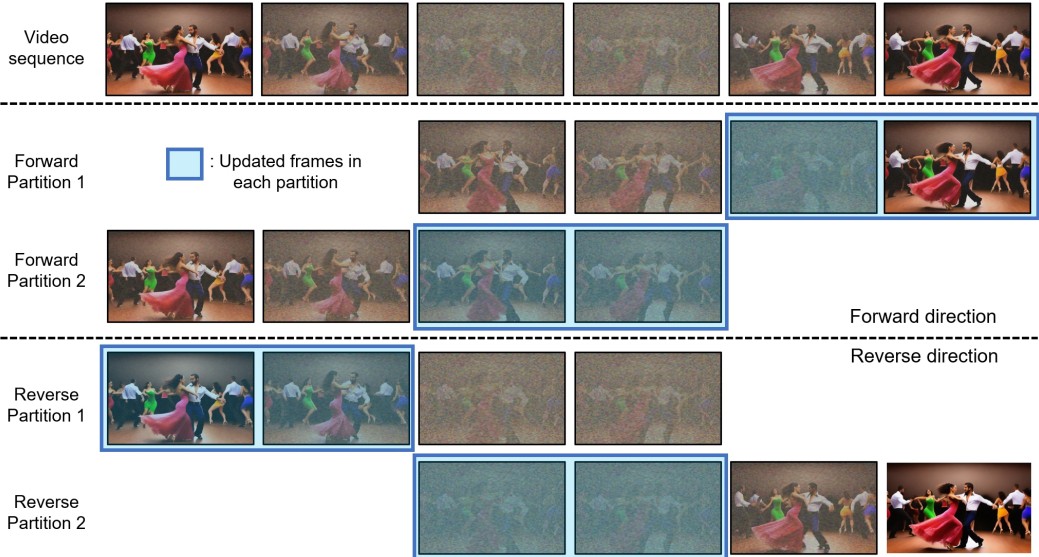

Figure 14: Lookahead denoising in both forward and reverse direction for bounded video generation. In our method, we apply the lookahead denoising only once for each denoising step, alternating the direction for each diffusion step.

denoising suggested in Section D, to denoise the sequence while considering the generation bias for both directions.

**Phase 3: Sequence shorter than model input size**
In this phase, lookahead denoising is no longer applied, as the sequence can be denoised without the latent partitioning. Generated frames are not removed from the sequence. Instead, they are input to the model along with the remaining noisy frames, to provide the model an additional information. The generated frames are not updated, as they already completed the diffusion generation process.

# D  Lookahead denoising

We utilized lookahead denoising along with the latent partitioning to make noise prediction be more accurate, as suggested in the previous method, FIFO. Specifically, as shown forward paritions in Fig. 14, the sequence is separated by several partitions to reduce the noise level difference between the intermediate frames. Moreover, lookahead denoising is utilize to further improve the accuracy of noise prediction, as the model predicts the noise, considering the cleaner frames.

For bounded video generation in our work, we utilize lookahead denoising in both the forward and reverse directions. Specifically, as illustrated in Fig. 14, we alternate the direction of lookahead denoising at each step of the diffusion process. Lookahead denoising impose the generation bias to align frames with the respect of direction of denoising. By leveraging both directions, our method improves the temporal consistency of the generated bounded video.

# E  Limitations and Potential negative impact

Our method preserves the context to generate coherent bounded video by the mixing strategy. Therefore, if there is big content difference between the bounds, the output video may presents degraded transition between the two sequences. For the social impact, as most of generative methods share, the video generation by the proposed work may induce a social disinformation by creating realistic fake videos. Moreover, there is risk of generating videos with harmful contents.

