# OpenReview forum: "Context Preserving Autoregressive Frame Generation for Bounded Video"
_NeurIPS.cc/2025/Conference — Submitted to NeurIPS 2025_

### Official Review · Reviewer_zqtm · 2025-06-19

**Clarity:** 2
**Significance:** 3
**Originality:** 2
**Rating:** 3
**Confidence:** 4

**Summary:**

This paper proposes a training-free method to interpolate between the starting and ending frames. Specifically, it uses uneven noise level to progressively predict the forward latent bidirectionally, and proposes a fusing mechanism for the two predicted frame sequences. It further adopts the SVGD guidance to improve generation quality. Qualitative experiments show that it can solve the problem more effectively, especially for the case with irreversible motion.

**Questions:**

Please refer to the Weakness part. Other questions/minor issues:
1. Can the authors clarify on the differences between bounded video generation and video interpolation (which may be more conventional) ?
2. Some notations are confusing. For example, $\tilde{z}_j$ in Algorithm 1 has not been explained.
3. Can the authors provide some failing cases? From Table 1 we can see that the proposed method underperforms some baselines on certain metrics. However, the qualitative comparisons only demonstrate the good case.

**Ethical Concerns:**

["NO or VERY MINOR ethics concerns only"]

**Final Justification:**

After reading the rebuttal, some of my concerns regarding weakness 1, 3 and 5 still stand. Therefore, I intend to keep my original score (borderline reject)

**Limitations:**

The base video generation model largely limits the generation quality as well as the length of the output video. To better assess the ability of the proposed method, I suggest the authors equipped it with more advanced (with more parameters) backbones.

**Paper Formatting Concerns:**

No.

**Quality:**

3

**Strengths And Weaknesses:**

Strengths:
1. The proposed method is straight-forward and easy to follow.
2. The qualitative experiment and the webpage show clear advantage of proposed method against baselines.
3. The proposed method can generate interpolated video of arbitrary length, rather than fixed length.

Weaknesses:
1. The technical contribution is marginal. The method is essentially a mixing of forward/backward generating sequence, with SVGD guidance. The method of adding more noise in the direction of generation has been studied in prior works [1].
2. Although the authors claim to avoid flipping the video frames to reduce generation bias, adding more noise in the direction of generation is also out of distribution for the base generative models, and this bias might affect generation quality for a training-free method. The authors should elaborate more on this.
3. The evaluation experiment seems problematic. For example, FIFO achieves the best score on Visual Quality, but why is the score of "Ours" bolded? Same case ViBiD on MAE(loop). The authors should provide a brief analysis on the under-performing metrics. On the other hand, human-evaluated metrics will be appreciated, which may be more reliable.
4. The adopted SVGD guidance seems independent of the main algorithm. The authors should equip the baselines with the same guidance, or remove the guidance to better analyze the performance gain against existing methods.
5. Compared to training-free methods, training-based methods may be more effective. And this paper lacks comparison with prior works such as [2][3][4][5]

References:

[1] Rolling Diffusion Models

[2] LDMVFI: Video Frame Interpolation with Latent Diffusion Models

[3] FILM: Frame Interpolation for Large Motion

[4] Animating Open-domain Images with Video Diffusion Priors

[5] Adapting Image-to-Video Models for Keyframe Interpolation

---

> ### Author Rebuttal · Authors · 2025-07-30
>
> We appreciate reviewer for the valuable comments.  Our responses are as follow:
>
> ### **W1. Contribution**
> *- Novelty issue regarding on autoregressive diffusion process*
>
> Our work focuses on bounded video generation which have unique challenge, in terms of causality between frames. Specifically, for open-ended video generation in previous works, next frame is predicted solely based on preceding frames.
> However, bounded video generation requires to consider not only the **past context**, but also the **future context** which is imposed by ending frames. If these two contexts are not properly integrated, discontinuity emerges in the output video due to the difference of the contexts.
>
> In this paper, we present a new problem **Context Gap**, and propose a novel method to guide autoregressive diffusion process to reduce the context discrepancy. Specifically, we leverage **contextual information** of a video sequence, which is not addressed in the prior work [1]. Moreover, our method is the first approach  to *capture* and *edit* the context of frame, which is novel for video generation domain.
>
> ---
>
> ### **W2. Analysis on bias**
> According to the reviewer's concern, we conducted an analysis for the bias of model. Following the previous work [6], we measure  **Relative MSE** to compare the model bias for two cases, flipped frames and reversed time schedule. Specifically, for predicted clean frames $\hat{z}(t)$ for forward direction video, we obtain the noise prediction using flipped $\hat{z}(t)$ to investigate the model's bias for flipped frames. For reversed time schedule, we utilized $\hat{z}(t)$ while the noise are added by the defined diffusion time. We present the comparison of *Relative MSE* in Table below.
>
> **Table 1** Related MSE for each diffusion time step (T=64)
> | Timestep  |0-7|8-15|16-23|24-31|32-39|40-47|48-55|56-63 |Average|
> |:------|---|---|---|---|---|---|---|---|---|
> |Flipped frame | 6.14 | 6.74 |5.35 |2.96 |1.82 | 1.39  | 1.25 | 1.03| 3.37 |
> |Reversed time schedule| 0.98  | 1.16  | 1.22  | 0.98  | 0.93  | 0.98 | 0.85 | 0.88 | 1.00 |
>
> The results shows that the mode bias is severe for flipped frames. Specifically, the error escalates  in the middle of diffusion generation process, where the context started to be formed. In contrast, the model exhibits relatively low bias for reversed time schedule across the whole time steps. The results verifies a superiority of our approach, avoiding the severe bias of model  against frames in flipped order.
>
> [6] Kim, Jihwan, et al. "Fifo-diffusion: Generating infinite videos from text without training." Advances in Neural Information Processing Systems, 2024
>
> ---
>
> ### **W3. Discussion on quantitative comparison**
>
> We will modify the table according to the comments. Moreover, according to the concerns of the reviewer, we briefly analyze the underperforming metrics. First, regarding on the visual quality, the main difference arises from the initialization of noisy new frames. Specifically, FIFO initialize the frames adding noise in the original video frame (start condition). However, our method initialize frames based on the predicted outputs of preceding frames near large diffusion time, where the predicted clean frames contain only low frequency components. Hence, the visual information provided in the initialization induces the slight gap of visual quality. We kindly remind the reviewer that FIFO is a method for open-ended video. Second, regarding on the MAE(loop), the ending condition is imposed near large diffusion time in the proposed bounded diagonal denoising. In other words, the first frame in the start context has smallest noise level, while the first frame in the ending context has largest noise level. This gap induces the slight differences in looped video.
>
> ---
>
> ### **W4. Issues  regarding on SVGD**
>
> *Relation of SVGD with main algorithm*
>
> We propose a SVGD-based guidance method to extract *contextual information* of video frames. Specifically, bounded video generation have unique challenge, in terms of causality between frames. Bounded video generation requires to consider not only the **past context**, but also the **future context** which is imposed by ending frames. If these two contexts are not properly integrated, discontinuity emerges in the output video due to the difference in the contexts.
> To address the context gap, we extract the SVGD-based context gradient, and guide the diffusion process to generate smooth transition between the two sequences.
>
> According to the reviewer's concern, we provide the ablation study on SVGD and mixing, which are the components to enhance the smooth transition between the two context. We compare temporal coherency in both pixel level and semantic level, using Temporal Flickering(pixel) and Background consistency(CLIP ViT-B/16) respectively. In particular, we measure the metric for frames located in 32-35 indices to focus on the context gap.  The baseline is naive application of bounded autoregressive denoising, without any techniques proposed in this paper. The results are presented in Table 2.
>
> **Table 2** Ablation study of each components for context gap. (32-35) refers the frame indices in the middle of the video.
> |    |Temporal Flickering (32-35) | Background consistency (32-35) | Temporal Flickering | Background consistency
> |------:|:---:|:---:|:---:|:---:|
> Baseline  | 0.856  | 0.925 | 0.934 | 0.932  |
> w/o mixing| 0.938  | 0.974  | 0.949  | 0.966  |
> w/o context gradient| 0.939  | 0.976  | 0.952  | 0.966  |
> Ours | 0.959 | 0.984 | 0.957 | 0.969 |
>
> The result demonstrates the effectiveness of each component in our method for the context gap. Without mixing and context gradient, the context gap is not fully addressed. Specifically, both metric evaluated in the middle of the video(33-35 frames) clearly shows the reduced discontinuity by the proposed components. In particular, temporal flickering is more sensitive to show the discontinuity by measuring the pixel-level difference, compared to the background consistency which measures the semantic continuity.
>
> ---
>
> #### **W5. Comparison with training-based method**
> We appreciate the reviewer's comments. Unfortunately, due to the limited time and resources, we could not conduct the comparison with training-based method.
>
>
> #### **Q1. Bounded Video Generation and Video Interpolation**
> We appreciate the reviewer's comment. Our work follows the task definition and experimental  settings suggested in the previous work [7]. Video interpolation focuses to generate frames to connect the boundary with shortest path. In contrast, bounded video generation aims to produce videos with plausible motions while accommodating varying degrees of contextual deviation.
>
> [7] Feng, Haiwen, et al. "Explorative inbetweening of time and space." ECCV, 2024.
>
> #### **Q2. $\tilde{z}_j$ in Algorithm 1**
> We denote the diffusion vector $z$ as $x$ for forward direction and $y$ for backward direction, as mentioned in the paragraph in page 5 of main paper. Accordingly, $\tilde{z}$ refers the vector for frame initialization obtained by equation (5) in manuscript. We will resolve the confusing notation in the final version.
>
> #### **Q3. Failure case**
> As discussed in W3, the bounded diagonal denoising induces timestep gap between the first frames of starting context and ending context. This induces slight discontinuity at the end of looping video, which is reflected in the MAE(loop).
>
> #### **Suggestion: Recent model**
> Our method is applicable only to video model that supports distinct time embedding for each frame. In case of Videocrafter2, the model duplicates the time embeddings for each frame, which enables distinct time condition for each frame. Recent models such as cogVideo(mmDiT), Wan/Hunyuan(DiT) do not support the distinct time embeddings, limiting compatibility with our approach. We also tried CausVid, however, the reversed frame generation is not supported, which is the key observation in section 3.1 in our paper. We believe it is due to the difference of model knowledge that is induced by the different training method, which would be future research direction.

---

> > ### Comment · Reviewer_zqtm · 2025-08-01
> >
> > I thank the authors for their detailed feedback. However, I still have some concerns:
> > 1. Can you kindly illustrate more on the difference between the concept of "bounded video generation" and "Video Interpolation"? Because I believe the latter is also required to consider not only the past context, but also the future context. Some models, like CogVideoX, can perform very well on this task, without additional fine-tuning.
> > 2. I don't think the authors have addressed Weakness 3 properly. The response is mainly on the difference between the proposed method and baselines. However, the question I have is that why the proposed method performs poorly on the metric Visual quality (both FVD_{16} and FID, against FIFO), and MAE(Loop) (against ViBiD), assuming that the data in Table 1 is valid. Also there are no manual metrics adopted.
> > 3. Unfortunately, I believe the comparison with recent methods such as CogVideoX is vital to prove the effectiveness of the method. Also, if this method can only be applied to a limited range of base video generation models (as has been indicated in the authors' last response), excluding the recent powerful backbones, I think the generalizability and impact of this method will be limited.

---

> ### Author Response · Authors · 2025-08-06
>
> We appreciate your thoughtful comments. Our responses are as follows:
>
> 1.
>
> Video interpolation aims to connect frames with shortest path. Specifically, in case of identical bound, video interpolation connects the two identical bounds with same frames, which is shortest path. In contrast, bounded video generation aims to generate video with plausible motions while satisfying the boundary. I believe that effectively managing context deviation within videos is a critical factor for bounded video generation.
>
> 2.
>
> We believe that the degradation in the mentioned metrics are due to the different initialization of the frames. Specifically, the mentioned methods provide more visual information to the frames. FIFO(FVD_{16}, VID) and ViBiD(MAE, Loop) initialize frames with original images which contains **high frequency** information. In contrast, our method use predicted frames which only includes **low frequency** information, to consider context. Accordingly, our method provide less prior information to the frames, which leads to the degradation of metric.
>
> Moreover, following the reviewer's comment, we provide human evaluation to verify a improvement in visual quality and the motion dynamics. 50 videos are evaluated by 22 participants. We compare our method with bounded video generation methods, TRF and ViBiD, while FIFO is excluded as it is an open-ended video. Preferences were rated on a scale of 0 to 5. The results demonstrate that our method improvements for both visual quality and motion dynamics.
>
> **Table 1** Human evaluation
> |  Methods  |Ours | ViBiD | TRF |
> |------:|---|---|---|
> |Visual quality| 3.624 | 2.857  | 2.460  |
> |Motion quality| 3.767 | 3.333  | 2.857  |

---

> > ### Author Response · Authors · 2025-08-09
> >
> > 3.
> >
> > Following the reviewer's comment, we conduct a comparison with the publicly released CogVideoX model finetuned for frame interpolation (comparison model: https://huggingface.co/feizhengcong/CogvideoX-Interpolation). Our method utilize the videocrafter 2 model, as CogVideoX architecture does not support autoregressive time schedule. The evaluation dataset is identical with the main paper. The results are shown in Table 2 and Table 3.
> >
> >  **Table 2** Comparison with cogVideoX for sinlge bound
> > |   |Dynamic Degree   | Temporal Flickering(unnormalized pixel MAE)  | MAE(loop)  |
> > |------:|---|---|---|
> > |Ours| 64.56\% | 10.847  | 0.076  |
> > |CogVideoX-finetuned| 1.94\% | 1.699  | 0.007  |
> >
> > Table 2 shows the comparison for looped video. We provide temporal flickering as unnormalized MAE between pixel values, to clarify the difference. Our method successfully generates video with dynamic motion, whereas finetuned CogVideoX model outputs static video without plausible motion.
> > Specifically, our method utilizes input context frames which provides the model with a motion prior, enabling rich motion dynamics. In contrast, CogVideoX relies on a single boundary frame which provides limited clue for movement. Accordingly, the CogVideoX connects the start and end frame with shortest path, replicating the boundary frame.
> >
> >
> >  **Table 3** Comparison with cogVideoX for two different bounds
> > |   |Dynamic Degree | MAE(start)  | MAE(end) |
> > |------:|---|---|---|
> > |Ours| 58.88\% | 0.097  | 0.101  |
> > |CogVideoX-finetuned| 26.90\% | 0.062  | 0.067 |
> >
> > Similar to looped video generation, Table 3 shows that CogVideoX exhibits a lower dynamic degree, connecting the two boundary frames via the shortest path. While CogVideoX is better to satisfy the boundary condition, the outputs often simply interpolate boundary frames, due to a limited motion clues. In contrast, our method outputs videos with dynamic movement using the motion prior from the input contexts.

---

### Official Review · Reviewer_wkyh · 2025-06-21

**Clarity:** 2
**Significance:** 1
**Originality:** 2
**Rating:** 2
**Confidence:** 4

**Summary:**

The authors propose a training free method that enables video generators to adhere to boundary conditions including, among others, last frame or first and last frame conditioning. To overcome limitations of previous methods that model reverse temporal dynamics, the authors propose a procedure, similar to "Diffusion Forcing" that progressively denoises video frames simultaneously from both video sides. Context-aware bidirectional autoregressive denoising and a mixing procedure are proposed to respect boundary conditions while limiting abrupt transitions in the middle of the videos.

**Questions:**

- To improve significance, the work could show how the current framework would compare to fine-tuning of a state-of-the-art open source video generator such as Wan or CogvideoX to respect boundary conditions. The work could show the quality gap between the two solutions, and training cost for the fine-tuning solution. If such analysis reveals a small gap in the quality of the attainable results between the two approaches, the saving of computational resources for training may increase the significance of the work. Unless such analysis is shown, significance of the work remains not sufficiently demonstrated.

**Ethical Concerns:**

["NO or VERY MINOR ethics concerns only"]

**Final Justification:**

The rebuttal did not address the main concerns raised in my initial review, which seem to be shared across some of the reviewers.

To raise above the bar of acceptance, I think the work should:
- Apply the method to state-of-the-art video generators
- Demonstrate a small performance gap between training-free and the widely adopted finetuning strategy
- Show with significative examples the practical appeal of the bounded generation task with respect to the widely used "frame-conditioned video generation" task, used in various flavors by many sota video generators.

In the absence of this I do not recommend acceptance of the manuscript.

**Limitations:**

The paper highlights limitations of the method.

**Paper Formatting Concerns:**

The paper would need a thorough proofreading to reach publication quality
- Spaces before reference blocks are used inconsistently
- Many typos are present. Some include LL 81, 82, 205, 209, Tab 1 caption, 221, 228

**Quality:**

2

**Strengths And Weaknesses:**

QUALITY
- While seemingly improving over similar frameworks as shown in Tab. 1, absolute quality of the results is low, with the produced videos presenting limited motion quality. Given the showcased level of obtainable quality, it is questionable whether the method could be used as an alternative to model fine-tuning.
- Incorrect bolding in Tab 1


CLARITY
- The framework is rather complex, requiring the usage of several tricks and hyper parameters. It is likely important for all such hyper parameters so be set up properly, but the paper does not provide enough insights or ablations showing the effects of each.

SIGNIFICANCE
- The paper addresses the problem of making a video generator adhere to given conditioning frames in a training free manner. This is a very specific value proposition from the work, whose value lies in the amount of training that can be saved to fine-tune large scale foundational video generation models to support such task natively. Performing such fine-tuning is, in practice, inexpensive, requiring only a few thousands fine-tuning iterations on limited amounts of computational resources. Such fine-tuning is thus inexpensive (especially compared to the initial cost of training a foundational video generator model) while producing high quality generators capable of respecting the boundary conditions. The proposed method is complex, requiring many hyper parameters, produces videos of limited quality and is not capable of handling the most interesting cases of boundary conditions with very different contexts.

ORIGINALITY
- Originality of the work appears fair. While drawing inspiration from frameworks such as Diffusion Forcing and mixing techniques common in training free long video generation to smooth model predictions, it combines them in an original manner.

---

> ### Author Rebuttal · Authors · 2025-07-30
>
> We appreciate the reviewer's comments. Unfortunately, due to the limited time and resources, we could not conduct the comparison with training-based method. Recent models such as Wan/Hunyuan(DiT) do not support the distinct time embeddings, which requires new architectural modification for the fine-tuning. Moreover, fine-tuning requires to construct text-video paired dataset, which was no able to conduct during limited rebuttal period.
>
> Regarding on errata, we will modify our manuscript in final version.
>
> We would like to emphasize the contribution of our method extends beyond the training.
> - We explore *context* of video frames, which is underexplored.
> - We propose method to *capture* and *edit* frame context, which is novel in video generation domain.
> - We present a novel problem *context gap*, and successfully mitigated by the proposed method.

---

> > ### Comment · Reviewer_wkyh · 2025-08-03
> >
> > I thank the authors for their rebuttal. My major concerns regarding the submission still stand after reading the rebuttal answers and other reviewer's comments that share my concerns on the timeliness of the submission and the significance of the task with respect to the traditional frame conditioned video generation.
> >
> > To summarize, I think the work should:
> > - Apply the method to state-of-the-art video generators
> > - Demonstrate a small performance gap between training-free and the widely adopted finetuning strategy
> > - Show with significative examples the practical appeal of the bounded generation task with respect to the widely used "frame-conditioned video generation" task, used in various flavors by many sota video generators.

---

### Official Review · Reviewer_PqNB · 2025-07-02

**Clarity:** 3
**Significance:** 2
**Originality:** 2
**Rating:** 3
**Confidence:** 4

**Summary:**

This paper proposes a novel bidirectional autoregressive method for generating videos that satisfy specific boundary constraints. By considering frame context and generating frames in both forward and reverse directions, the method addresses the unnatural motion often produced by traditional approaches when handling irreversible motions. It introduces a context-aware bidirectional denoising technique, as well as frame initialization strategies and context gradient mixing methods, to bridge the context gap between the two directions and generate coherent video sequences. Experimental results demonstrate that the proposed method produces videos with more realistic motion dynamics and higher visual quality than existing approaches, particularly when adhering to boundary constraints.

**Questions:**

1) Although the authors have demonstrated the advantages of the proposed method in two tasks, the evaluation of video generation under different scenarios remains insufficiently comprehensive. Beyond the reported cases, did the authors assess other video lengths? Is the method's performance stable across a broader range of video durations? Is it accurate in more complex scenes? It is recommended that the authors include additional experimental results on videos of varying lengths to more convincingly demonstrate the robustness and generalization capability of the method.
2) While the ablation studies support the effectiveness of the proposed method in bridging context gaps, the analysis of the root causes of these gaps and the method’s sensitivity to such gaps under different scenarios remains limited. The authors are encouraged to further investigate the mechanisms behind the formation of context gaps and to evaluate the method’s performance under varying degrees of context mismatch, in order to better understand its strengths and limitations.
8. Limitations

**Ethical Concerns:**

["NO or VERY MINOR ethics concerns only"]

**Final Justification:**

I appreciate the authors' rebuttal, which partially resolved my concerns. After reviewing all feedback, I maintain my original opinion.

**Limitations:**

yes

**Paper Formatting Concerns:**

The paper's formatting basically meets the requirements.

**Quality:**

3

**Strengths And Weaknesses:**

**Strengths**
1) This paper proposes a novel bidirectional autoregressive method for bounded video generation. By using context-aware bidirectional denoising, along with frame initialization and context mixing strategies to reduce context gaps, it effectively addresses coherence issues between forward and backward frame generation. These designs help mitigate the forward-generation bias commonly seen in video models, thereby enhancing the realism and dynamics of the generated videos.
2) The proposed method outperforms existing approaches like FIFO, TRF, and ViBiDSampler in terms of visual quality, temporal consistency, and boundary satisfaction. For example, under both identical and distinct boundary conditions, the generated videos demonstrate richer visual details and more natural motion patterns, including fluid dynamics like water splashes.
3) The paper conducts ablation studies on irreversible motion generation, video length impacts on context, and context gaps. These studies validate the effectiveness of each component of the proposed method, offering deeper insights for future research.

**Weaknesses**
1) Although the paper presents a bidirectional autoregressive method tailored for bounded video generation, the core ideas are built upon existing autoregressive and diffusion-based generative models [1] [2]. For instance, techniques such as Stein Variational Gradient Descent (SVGD) have already been explored in generative modeling [3], including for video data. While the authors introduce a novel integration and application of these ideas, the conceptual innovation appears incremental rather than fundamentally novel.

2) The experimental setup may limit a comprehensive evaluation of the method’s performance. The datasets used primarily consist of short video clips with specific irreversible motions, which may not reflect performance on longer or more complex motion sequences. Furthermore, the chosen evaluation metrics may not sufficiently capture aspects such as semantic consistency or the richness of visual details, potentially underrepresenting the quality and diversity of the generated content.
3) The paper lacks comprehensive comparisons with recent related work, which hinders a clear assessment of the method's distinct contributions and novelty in the current research landscape. Additionally, the robustness of the approach under varying motion complexities, boundary conditions, and noise disturbances is not thoroughly evaluated. This insufficient validation makes it challenging to assess the method’s reliability in diverse real-world scenarios.
4) In Table 1 of the paper, the metric labeled 'Visual quality' is one where lower values indicate better performance. However, the proposed method is not the best according to this metric, yet it is still highlighted in bold as if it were the best.



[1] Weijie Kong, Qi Tian, Zijian Zhang, Rox Min, Zuozhuo Dai, Jin Zhou, Jiangfeng Xiong, Xin Li, Bo Wu, Jianwei Zhang, et al. Hunyuanvideo: A systematic framework for large video generative models.

[2] Bin Lin, Yunyang Ge, Xinhua Cheng, Zongjian Li, Bin Zhu, Shaodong Wang, Xianyi He, Yang Ye, Shenghai Yuan, Liuhan Chen, et al. Open-sora plan: Open-source large video generation model.

[3] Kim, Subin, et al. "Collaborative score distillation for consistent visual synthesis."

---

> ### Author Rebuttal · Authors · 2025-07-30
>
> We appreciate reviewer for the valuable comments.  Our responses are as follow:
>
> ### **W1. Contribution**
> *1. Novelty issue regarding on autoregressive diffusion process*
>
> Our work focuses on bounded video generation which have unique challenge, in terms of causality between frames. Specifically, for open-ended video generation in previous works, next frame is predicted solely based on preceding frames.
>
> In contrast, bounded video generation requires to consider not only the **past context**, but also the **future context** which is imposed by ending frames. If these two contexts are not properly integrated, discontinuity emerges in the output video due to the difference of the contexts. In this paper, we present a new problem **Context Gap**, and propose a novel method to guide autoregressive diffusion process to reduce the context discrepancy.
>
> *2. Novelty issue on SVGD*
>
> We propose a SVGD-based diffusion guidance method to extract contextual information of video frames, which is fundamentally different from prior work~[3] in both motivation and mechanism. The primary goal of [3] is to preserve inter-frame relationships after video editing, where the authors employ a Gaussian kernel based on the L2 difference between frames. This approach relies solely on **visual similarity**, without considering the temporal causality between frames.
>
> In contrast, our method mainly aims to capture **contextual information** of video frames. To consider temporal causality, our method presents a moving average instead of gaussian kernel. The moving average monotonically decreases with respect to the **distance in frame index**, which reflects temporal dependency between frames. Overall, our paper focuses on the emergence of frame context and propose a novel method to adjust context for bounded video generation, which is underexplored in previous works.
>
> ---
>
> ### **W2. Evaluation on semantic consistency**
> We kindly remind the reviewer that the prompts are randomly generated by LLM, without consideration on an irreversibility of motions. Regarding on the metric, temporal consistency is evaluated in both pixel level and semantic level consistency. Specifically, *"Background consistency"* measures cosine similarity between CLIP visual features(CLIP ViT-B/32), while *"Temporal flickering"* measures MAE difference of pixel values. According to reviewer’s concern, we provide `Subject Consistency’ which measures cosine similarity between [CLS] token features of DINO. In Table 2, the result shows that our method outperforms the previous methods in *Subject Consistency*, by improved semantic consistency. Moreover, we add *Imaging Quality* of VBench, to additionally evaluate visual quality of videos. Our method also outperforms the previous approach in both metrics.
>
> |   | Ours | FIFO | ViBiD | TRF |
> |:------:|---|---|---|---|
> |Subject Consistency| 0.916 | 0.901  | 0.906  | 0.915  |
> | Imaging Quality | 62.73% | 62.85% | 61.27% | 60.18%|
> ---
>
> ### **W4. Errata**
> We will modify the main paper in final version
>
> ---
>
> ### **(W3/Q1/Q2). Further analysis**
>
> ### *1. Stability analysis*
>
> Following the reviewer's comment, we provide stability analysis of our method for the variable length of video. Specifically, we focus on the visual quality and context deviation, which is directly affected by the video length. For the measurement of context gap, we obtain the MAE difference between the frames in the transition of the two sequences(MAE between 33,34 frames). In Table 2, we provide FID and context gap for various video length.
>
> **Table 2** Ablation study for variable length
> |  Length  |50|100|120|150|170|
> |:------:|---|---|---|---|---|
> |Temporal Flickering| 0.957 | 0.961  | 0.963  | 0.964  | 0.966 |
> |Context Gap (MAE in pixel)| 10.53 | 7.99 | 8.23 | 7.16  | 7.06 |
> |FID| 31.61  |31.217  | 30.419 | 30.049 | 30.604 |
>
> As shown in Table 2, our method retains the performance in varying video length, in terms of both temporal coherency and visual quality.
>
>
> ### *2. Further study on context  gap*
>
> *2.1. Varying context mismatch*
>
> According to the reviewer’s concern, we investigate the performance of our method to the varying degree of context gap. To quantify the context gap for a given motion, we independently extended input frames using bidirectional denoising, and calculated the MAE between the final frames of each direction. Then, we measure the MAE difference in the middle of video generated by our method to the corresponding motion. A scatter plot would be appropriate for this analysis. However, considering the conference guideline, we present statistical values instead. We grouped the instances into 7 distinct partitions according to the context gap of original motion. Then, we measure mean and standard deviation of each groups, as shown in Table 3.
>
> **Table 3** Varying degree of context gap. Context Gap is unnormalized MAE difference in pixel values.
> |  Percentile (\%)  |  Partition 1 (~16.67) |  Partition 2 (~33.33)  | Partition 3 (~50) | Partition 4 (~66.67)  | Partition 5 (~75) |Partition 6 (~83.33) | Partition 7 (~100) |
> |:------:|---|---|---|---|---|---|---|
> | Given motion (Context Gap) | 25.81 $\pm$6.06  | 35.67 $\pm$2.08 | 41.07 $\pm$1.52 |45.51$\pm$ 1.25 | 51.48 $\pm$1.61 |  56.80$\pm$ 1.68 | 65.23$\pm$ 4.55 |
> |Ours ( Context Gap )|  6.93 $\pm$ 2.38  |   9.25 $\pm$ 4.06  |  9.14 $\pm$ 3.76  | 11.11 $\pm$ 4.22 | 11.77$\pm$ 5.03  | 11.68 $\pm$ 4.97 | 14.32 $\pm$ 7.76|
>
> The result shows that our method improve the gap for all groups. However, in the partition 7, the result shows increased mean and deviation, which indicates the presence of discontinuous video in our result.
>
>
> *2.2. Formation of frame context and context gap*
>
> We appreciate the valuable comment. The context gap emerges by the causality of frames which is unidirectional. The next frame is generated only considering the past context, not the future context. Therefore, the context of frame deviates as the dependency to the initial frame become weak, which is also demonstrated in section 4.3.2 of main paper. According to the causal structure of frames, the video model has bias of generating frames in forward direction. We believe the the model's bias for the unidirectional frame forms the context gap in video generation. To support our claim, we present the bias analysis for video model.
>
> Following the previous work [4], we measure  **Relative MSE** to compare the model bias for two cases, flipped frames and reversed time schedule. First, we investigate the model's response for flipped frame (reversed causality). Next, we explore the model's output for reversed time scheduling with forward frame, which demonstrates the potential of our approach to indirectly extract the reversed causality from model's knowledge.
>
> Specifically, for predicted clean frames $\hat{z}(t)$ for forward frames, we obtain the noise prediction using flipped $\hat{z}(t)$. For reversed time schedule, we utilized $\hat{z}(t)$ while the noise are added by the reversed diffusion time. We present the result of *Relative MSE* in Table 4.
>
> **Table 4** Average of related MSE for each diffusion time steps
> | Timestep  |0-7|8-15|16-23|24-31|32-39|40-47|48-55|56-63 |Average|
> |:------|---|---|---|---|---|---|---|---|---|
> |Flipped frame | 6.14 | 6.74 |5.35 |2.96 |1.82 | 1.39  | 1.25 | 1.03| 3.37 |
> |Reversed time schedule| 0.98  | 1.16  | 1.22  | 0.98  | 0.93  | 0.98 | 0.85 | 0.88 | 1.00 |
>
> The results shows that the mode bias is severe for flipped frames. Specifically, the error starts to escalate from the middle of diffusion generation process, which indicates the formation of context. Simultaneously, the prediction of model contains the change of contents, which is demonstrated by high value of *Related MSE*. The analysis shows the unidirectional causal structure within the model, which causes the context gap.
>
> Moreover, the model exhibits relatively low bias for reversed time schedule across the whole time steps. The results verifies a potential of our approach to indirectly extract reversed causality, leveraging the unidirectional bias of video model.
>
> [4] Kim, Jihwan, et al. "Fifo-diffusion: Generating infinite videos from text without training." Advances in Neural Information Processing Systems, 2024

---

> > ### Comment · Reviewer_PqNB · 2025-08-08
> >
> > I appreciate the authors' rebuttal, which partially resolved my concerns. After reviewing all feedback, I maintain my original opinion.

---

> ### Comment · Area_Chair_BnDC · 2025-08-06
> **Comment by AC**
>
> Dear Reviewer PqNB,
>
> Thank you for your participation in the review process. Please engage in the discussion phase by following these guidelines:
>
> - Read the author rebuttal;
> - Engage in discussions;
> - Fill out the "Final Justification" text box and update the "Rating" accordingly.
>
> The deadline is Aug 8, 11.59pm AoE.
>
> Thanks,
>
> AC

---

### Official Review · Reviewer_jm1c · 2025-07-03

**Clarity:** 2
**Significance:** 2
**Originality:** 3
**Rating:** 4
**Confidence:** 3

**Summary:**

The paper targets bounded video generation—producing a finite video that must respect given start and/or end frames—and observes that existing diffusion‑based methods typically create the reverse segment by simply time‑flipping forward‑generated frames, an approach that clashes with the innate forward‑sampling bias of current video diffusion models and yields unrealistic motion for irreversible actions. To solve this, the authors propose a training‑free, bidirectional autoregressive framework that denoises frames progressively in both forward and backward directions using a “reversed‑diagonal” timestep schedule, thereby generating the reverse sequence without flipping. They introduce a context‑aware gradient (inspired by Stein Variational Gradient Descent) that lets each frame share denoising updates with its neighbors and a global mixing strategy that blends latent predictions from the two directions to eliminate the mid‑sequence discontinuity (“context gap”). A three‑phase inference schedule alternates look‑ahead denoising while keeping the model’s input length constant, avoiding the length mismatch that degrades prior single‑pass methods. Experiments on VideoCrafter‑2 across 400 GPT‑generated prompts show that, compared with FIFO (open‑ended) and flip‑based bounded video baselines (TRF, ViBiDSampler), the proposed method achieves comparable or better temporal consistency (VBench flicker/background scores), better perceptual quality (lower FVD/FID), more natural irreversible motions, and similar boundary adherence, all without extra training.

**Questions:**

N/A

**Ethical Concerns:**

["NO or VERY MINOR ethics concerns only"]

**Final Justification:**

I do not have many concerns with the paper initially and would keep my positive vote.

**Quality:**

2

**Strengths And Weaknesses:**

Strength:

-- Bounded generation is important for interpolation / completion but under‑studied, and the flip‑based limitation is well explained.

-- The proposed reversed‑diagonal trick to sample backward without re‑training is simple and effective.

-- Context‑gradient idea provides a principled way to let neighboring frames communicate; ablations indicate gains.

-- Qualitative evidence on irreversible motions convincingly shows the benefit over flip methods.

Weakness:

The biggest concern I have for this paper is its timelyness -- it still uses VideoCrafter 2 as its backbone, and VideoCrafter 2 itself is known for having low FPS and strong artifacts -- it's 1.5 years old already. Concurrent work that share similar ideas, such as FramePack, already started doing experiments on Wan/Hunyuan. The authors show some comparisons on their website, but firstly there are not that many, secondly it is difficult to argue about the quality gain, the examples all exhibit strong artifacts...

---

> ### Author Rebuttal · Authors · 2025-07-30
>
> We appreciate the fruitful comments. The response is as below:
>
> ### **Recent models**
>
> Our method is applicable only to video model that supports distinct time embedding for each frame. Specifically, *Videocrafter2* model duplicates the time embeddings to the dimension of frame length, which enables distinct time condition for each frame. Recent models such as cogVideo(mmDiT), Wan/Hunyuan(DiT) do not support the distinct time embeddings, limiting compatibility with our approach. We also tried CausVid, however, the reversed frame generation is not supported, which is the key observation in section 3.1 in our paper. We believe it is due to the difference of model knowledge that is induced by the different training method, which would be future research direction.
>
> ---
>
> ### **Evidence for reversed frame generation**
>
> To support the reversed frame generation in the selected model, we conduct an analysis on the model’s prediction. Following the previous work[1], we present the *Relative MSE*, as shown in Table 1.
>
> **Table 1** Related MSE for each diffusion time step ($T$=64)
> | Timestep  |0-7|8-15|16-23|24-31|32-39|40-47|48-55|56-63 |Average|
> |:------|---|---|---|---|---|---|---|---|---|
> |Flipped frame | 6.14 | 6.74 |5.35 |2.96 |1.82 | 1.39  | 1.25 | 1.03| 3.37 |
> |Reversed time schedule| 0.98  | 1.16  | 1.22  | 0.98  | 0.93  | 0.98 | 0.85 | 0.88 | 1.00
>
> The results demonstrates that the reversed frame generation is supported, showing low relative MSE. In contrast, the value is high for flipped frame, which verifies the severe bias for the prediction on flipped frames. Specifically, the error in flipped frame starts to escalate from the middle of diffusion generation process, which indicates the formation of frame context.

---

> > ### Comment · Reviewer_jm1c · 2025-08-09
> >
> > I do not have many concerns with the paper initially and would keep my positive vote.

---

### Note · Authors · 2025-08-14

We appreciate reviewer’s constructive comments and discussions on this work. To clarify the contribution, we highlight the key research statement addressed in this work, which are underexplored in video diffusion generation:

-	How can **contextual information** of frames be effectively extracted?
- How does video model's bias affects to generate physically plausible motions?
-	How can **past** and **future** contexts be utilized/edited to generate coherent bounded videos?

To achieve the research goal, we proposed context-aware diffusion guidance that mitigates a proposed novel problem, **Context Gap**.

During the rebuttal, the main concern was compatibility with more recent models, as our method is based on a model released in last year. While recent pretrained models currently do not support reversed frame generation, we view this as a promising future direction for video model development, which lies beyond the scope of the presented work.

---

### Decision · Program_Chairs · 2025-09-17

**Decision:**

Reject

**Comment:**

The paper introduces a training-free approach to generate videos that satisfy specific boundary constraints by progressively denoising the video frames from two sides. After the rebuttal, three reviewers maintained negative ratings while one maintained a positive rating. The reviewers highlight that the studied problem is important since the fine-tuning cost can be saved, the method can generate videos of arbitrary length, and the proposed bidirectional autoregressive method is novel.

However, some concerns remain unsolved. Specifically, the paper does not include comparisons with recent work, and reviewers suggest applying the proposed approach to state-of-the-art video generation models. Also, the performance gap between the proposed training-free approach and fine-tuning strategies should be well studied. Without such an analysis, it is non-trivial to understand the value of this work over existing fine-tuning methods. Reviewers also suggest comparisons with the "frame-conditioned video generation" task that is used in various video generation models.

Overall, the paper is recommended for rejection, and the authors are encouraged to incorporate the feedback from reviewers.